# SAM-MIL: A Spatial Contextual Aware Multiple Instance Learning Approach for Whole Slide Image Classification

## ABSTRACT

Multiple Instance Learning (MIL) represents the predominant framework in Whole Slide Image (WSI) classification, covering aspects such as sub-typing, diagnosis, and beyond. Current MIL models predominantly rely on instance-level features derived from pretrained models such as ResNet. These models segment each WSI into independent patches and extract features from these local patches, leading to a significant loss of global spatial context and restricting the model's focus to merely local features. To address this issue, we propose a novel MIL framework, named SAM-MIL, that emphasizes spatial contextual awareness and explicitly incorporates spatial context by extracting comprehensive, image-level information. The **S**egment **A**nything **M**odel (SAM) represents a pioneering visual segmentation foundational model that can capture segmentation features without the need for additional fine-tuning, rendering it an outstanding tool for extracting spatial context directly from raw WSIs. Our approach includes the design of group feature extraction based on spatial context and a SAM-Guided Group Masking strategy to mitigate class imbalance issues. We implement a dynamic mask ratio for different segmentation categories and supplement these with representative group features of categories. Moreover, SAM-MIL divides instances to generate additional pseudo-bags, thereby augmenting the training set, and introduces consistency of spatial context across pseudo-bags to further enhance the model's performance. Experimental results on the CAMELYON-16 and TCGA Lung Cancer datasets demonstrate that our proposed SAM-MIL model outperforms existing mainstream methods in WSIs classification.

## CCS CONCEPTS

• **Computing methodologies** → *Object identification*; *Computer vision tasks*; *Object recognition*.

## KEYWORDS

Whole Slide Image Classification, Deep Learning, Multiple Instance Learning, Weakly Supervised Learning

## 1 INTRODUCTION

The analysis of histopathological images is pivotal in modern medicine, particularly in cancer treatment, where it is considered the gold standard for diagnosis [21, 25, 31, 49]. The digitization of pathological

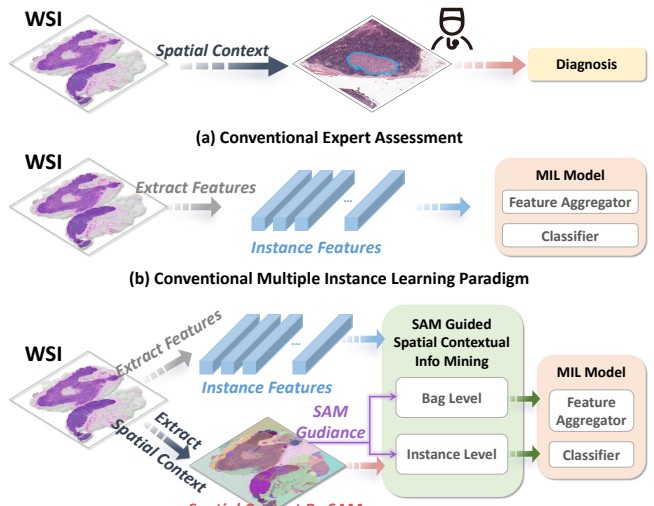

**(a) Conventional Expert Assessment**

**(b) Conventional Multiple Instance Learning Paradigm**

**(c) Proposed MIL Paradigm with SAM Gudiance**

**Figure 1: Top: The conventional pathologists' assessment strongly relies on the spatial contextual features in the WSI. Middle: The conventional MIL paradigm relies solely on individual features, overlooking the global spatial context between patches. Bottom: The proposed MIL paradigm introduces the SAM, which utilizes the spatial context and guides the optimization of the MIL model.**

images into Whole Slide Images (WSIs) via digital slide scanners has opened new avenues for computer-assisted analysis [8, 34]. Given the significant dimensions of WSIs and the absence of detailed pixel annotations, the analysis of histopathological images is commonly approached via Multiple Instance Learning (MIL) [10, 29, 36], where MIL serves as a form of weakly supervised learning [26, 41]. In the MIL framework, each WSI or slide is considered as a bag, where each slide is divided into thousands of separate patches through a patching operation. A pre-trained feature extractor [17, 27, 34, 46] is used to extract features from each patch. The extracted instance features are stored within the bag as unmarked instances (patches) to serve as input data. A bag is classified as positive if it contains at least one instance indicative of disease; otherwise, it is classified as negative.

While the MIL approach simplifies the problem by segmenting images into small patches for independent feature extraction [16, 17, 27, 34, 46], this method fundamentally overlooks the global spatial context among patches. Since each patch is treated independently with features extracted in isolation, the MIL model tends to focus solely on local features, as illustrated in Figure 1(b). The absence of this spatial context, especially in the ultra-high resolution context of WSIs, is detrimental to understanding tissue architecture, identifying pathological changes, and classifying diseases. For instance, certain pathological features may span multiple patches and

can only be correctly identified and interpreted when considering the spatial relationship between these patches. Pathologists, when analyzing WSIs, consider the spatial layout of tissues and the relationships between adjacent patches to identify tumor markers and understand pathological processes, as depicted in Figure 1(a). Previous studies have explored the modeling of spatial context through implicit methods. These approaches primarily involve constructing models capable of capturing long-range dependencies within images [1, 34], as well as utilizing the spatial coordinates to form graph networks [6]. Nevertheless, these methods often fail to take full advantage of the richness of spatial contextual information. This implicit use of spatial context may underestimate the critical role of spatial relationships in tissue architecture analysis and tumor diagnosis.

However, there has been insufficient focus on explicitly recovering and utilizing the spatial context lost during the patching process. To effectively integrate spatial contextual information into MIL, it is essential to leverage raw image-level features. As a pre-trained foundation model, the **S**egment **A**nything **M**odel (SAM) [19] can be directly applied to WSI analysis, providing additional, semantic-independent visual segmentation prior features without the need for further training or fine-tuning. The SAM provides high-quality segmentation features from slides, thereby making it an outstanding tool for extracting spatial context from raw WSIs.

In this paper, we propose a novel WSI classification method named SAM-MIL, which aims to mine and exploit spatial context information from slides to enhance the MIL model. As a foundational segmentation model, SAM acquires and encodes prior visual context knowledge in spatial and appearance aspects. By deducing these knowledge in SAM, the original slide is segmented, thereby explicitly modeling the relationships of instances based on neighboring relations and visual similarities, termed 'spatial context'. Utilizing this contextual knowledge, we elaborate a SAM-Guided Group Masking ($SG^2M$) scheme that filters out the redundant instances by segments according to their areas to alleviate the extreme distribution imbalance of instances. To reduce the typical feature loss risk caused by the sharp instance masking ratio, we aggregate the instances by segment categories and complement them into the preserved instances for introducing a completed representation. Furthermore, SAM-MIL divides instances to create additional pseudo-bags, enhancing the training set, while ensuring spatial context consistency across pseudo-bags to further improve the model. Extensive experiments on several popular benchmarks demonstrate the superiority of SAM-MIL over baselines, and also confirms the importance of spatial contextual information in WSI classification. The contribution of this paper is summarized as follows:

- We propose a meticulously MIL framework based on spatial contextual awareness named SAM-MIL. We pioneer the use of SAM to extract spatial contexts in WSIs and explicitly integrate these spatial contexts into MIL model training. Validated using the CAMELYON-16 and TCGA Lung Cancer datasets, our model demonstrates superior classification performance relative to mainstream MIL methods and confirms the significance of incorporating spatial context in WSI classification.
- We design a masking strategy termed $SG^2M$ and a global group feature extractor to mitigate the class imbalance. We

dynamically assign mask ratios to different categories by grouping independently extracted instance features based on their spatial contexts. Meanwhile, a global group feature extractor is employed to compensate for the potential risk of typical feature loss.
- We design a SAM spatial context-based consistency constraint and pseudo-bag spliting strategy to supplement the limited training data. Instances are split during training to generate additional pseudo-bags. Spatial context guides model training through consistency loss, ensuring the consistency of same-category instances in pseudo-bags.

## 2 RELATED WORK

### 2.1 Multiple Instance Learning in WSI Analysis

Multiple Instance Learning (MIL) [10] represents the most widely applied paradigm in WSI analysis [3, 17, 27]. Given the ultra-high resolution of WSIs, instance features are typically extracted using pre-trained models [4, 16, 20, 27, 33, 50]. Feature extraction from local instances eliminates the spatial context in the original WSIs, and the extracted features only contain localized information in the delineated patches, which is used for subsequent bag label prediction. Previous algorithms can be broadly categorized into two types: instance-level [3, 11, 14, 44] and embedding-level [8, 35, 42, 43, 46]. The former obtains instance labels and aggregates them to obtain bag labels, while the latter aggregates all instance features into a high-level bag embedding for bag prediction. However, the MIL model described previously did not focus on how to recover and utilize the spatial context lost during the patching process. Instead, it used the patch features directly for training. Some studies are attempting to reconstruct spatial context. For example, Chen et al. [6] conceptualizes whole slide images as 2D point clouds, employing Patch-based Graph Convolutional Networks to foster context-aware survival predictions by leveraging the spatial relationships between patches to learn context-aware embeddings implicitly. Bai et al. [1] introduces a novel framework utilizing transformers for object localization, implicitly capturing spatial context through activation diffusion techniques. But the reconstruction of spatial context in current MIL models is primarily done implicitly. In this paper, we aim to provide explicit spatial context to guide the MIL model training by introducing visual features at the image level to improve the performance of WSIs classification.

### 2.2 SAM in Medical Image Analysis

Foundation models have profoundly revolutionized traditional artificial intelligence across various domains, including medicine [5, 24], due to their exceptional zero-shot and few-shot generalization capabilities in downstream tasks [22, 40]. Among these, the Segment Anything Model (SAM) [19], as a pioneering image segmentation foundation model, has garnered widespread attention for its ability to generate accurate target masks in a fully automatic or interactive manner. This marks the entry of the prompt-driven paradigm into the realm of image segmentation, achieving favorable outcomes in numerous tasks [7, 12, 28, 39, 45]. However, due to the uniqueness of medical images (with traditional foundation models predominantly trained on non-medical datasets), the effectiveness of SAM in medical imaging still requires continuous exploration.

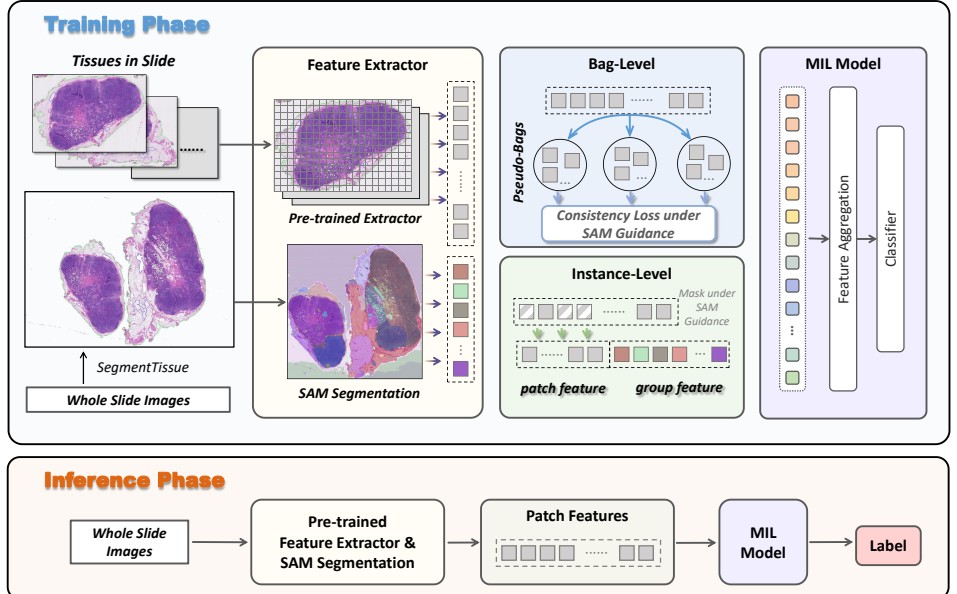

**Figure 2: Overview of the proposed SAM-Guided WSI classification model. In the Feature Extractor stage, WSI slides are segmented into corresponding tissues. Following the patching operation, each tissue sequentially extracts features from each patch. Simultaneously, SAM performs segmentation on the entire slide, extracting representative features from each region as group features based on spatial context. In the Feature Aggregation stage, the spatial context of SAM is utilized at two levels. At the Instance level, instance grouping masks are applied under the guidance of SAM, while at the Bag level, pseudo-bag training loss and consistency loss are calculated under SAM's guidance to constrain the model's training. This approach ensures that both detailed instance-level features and the broader bag-level insights contribute to the model's learning process.**

Nevertheless, extensive work has already introduced SAM into the field of medical images, mainly focusing on lower-resolution image segmentation tasks [15, 18, 30, 32, 38, 47]. This work can be roughly categorized by medical imaging modalities: 1) CT images [15, 18, 32], 2) MRI images [30, 47], 3) Endoscopic images [38], etc. However, much of this work remains a step behind the current mainstream methods, requiring fine-tuning of the SAM model or the provision of suitable prompts. Hence, the direct application of SAM to medical imaging tasks is limited in generalizability, with significant differences across various datasets and tasks [48]. Due to the ultra-high resolution of WSI images and the extremely small proportion of diseased areas, there have been few attempts to introduce SAM. Deng et al. [9] evaluated SAM for tumor segmentation, non-tumor tissue segmentation, and cell nucleus segmentation in WSIs data, demonstrating the feasibility of SAM application in WSIs. Beyond this, there is virtually no work involving SAM in the WSIs domain. Previous work has demonstrated the generalizability of SAM and its usefulness in medical images, but due to the specificity of medical images, there are still substantial challenges for SAM to be used directly as a medical image segmentation model. In this paper, we explore the adaptation of the original SAM to WSIs classification tasks. Through ingenious model design, the spatial context mined by SAM is utilized to guide the training of MIL models, offering a novel approach for the application of SAM in medical images.

## 3 PROPOSED METHOD

### 3.1 Preliminary

Within the paradigm of MIL, we encounter the scenario where a data collection is comprised of bags, denoted as $\mathbf{B} = \{\mathbf{B_1}, \mathbf{B_2}, ..., \mathbf{B_N}\}$, each consisting of a number of instances $\mathbf{B_i} = \{x_1^i, x_2^i, ..., x_n^i\}$ in a $D$-dimensional feature space. In classification tasks, a bag is associated with a predetermined label $Y$, while each instance within the bag carries an unspecified label $y_i$. The bag is classified as positive if it contains at least one positive instance; if not, it is deemed negative. The learning objective of a MIL model $\mathcal{M}$ is to infer the bag label considering the information embedded within its instances, formalized as:

$$\hat{Y} \leftarrow \mathcal{M}(\mathbf{B_i}) := C\left(\mathcal{A}\left(\{\mathcal{F}(x_j^i)|j = 1, ..., n\}\right)\right), \quad (1)$$

where $\mathcal{F}$ denotes the feature extraction function, transforming each instance into a more expressive feature representation; $\mathcal{A}$ is the feature aggregation function that synthesizes the extracted features into a bag representation; and $C$ is the classification function, predicting the bag label from its aggregated feature representation.

**However, traditional MIL paradigms primarily concentrate on the local features of patches, resulting in a substantial overlook of spatial context.** In this paper, we will explicitly introduce spatial context for guiding model training via SAM in the feature extraction phase and the feature aggregation phase:

$$\hat{Y} \leftarrow \mathcal{M}_{SAM}(\mathbf{B_i}) := C_{SAM}\left(\mathcal{A}_{SAM}\left(\{\mathcal{F}_{SAM}(x_j^i)|j = 1, ..., n\}\right)\right), \quad (2)$$

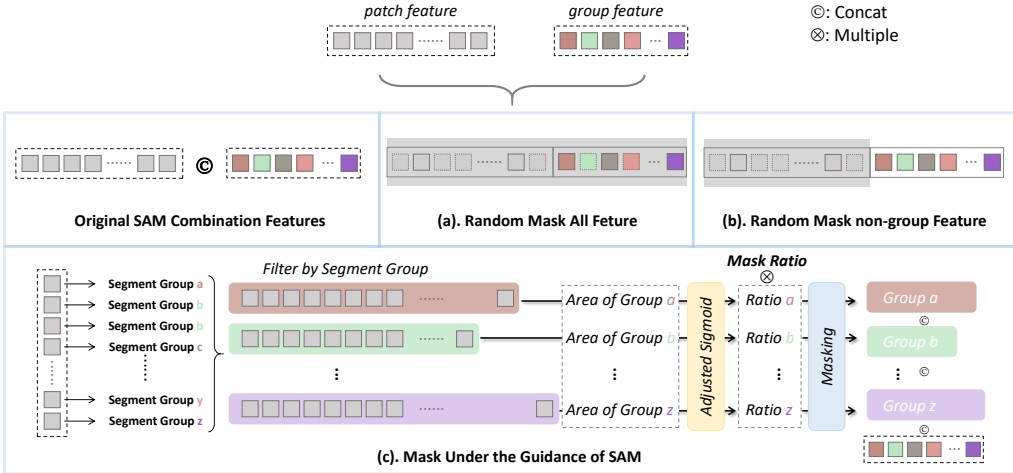

**Figure 3: Illustration of proposed masking strategy. We propose three masking strategies for instances. The first two strategies involve randomized masks. The third strategy is our proposed spatial context-based SAM-Guided Group Masking (SG²M), which groups various SAM segmentation categories and enforces a dynamic mask ratio within each group.**

where $SAM$ denotes the method that is guided by the spatial context within the Segment Anything Model.

## 3.2 Spatial Contextual Aware WSI Classification

As illustrated in Figure 2, we introduce a novel approach to the MIL model that enhances classification performance by incorporating spatial context. Previous works predominantly exploited local features from WSIs, overlooking the potential of global visual content. To address this, we integrate spatial context from the SAM into our MIL framework. **Specifically, we have designed a composite feature extractor that not only includes the traditional feature extraction phase but also leverages the segmentation results from SAM to extract a representative group feature from each segmented region.** Let $S = \{s_1, s_2, \ldots, s_l\}$ denote the output of the SAM, representing $l$ segmented areas. For each segmented area $s_r$, the composite feature $g_r$ is extracted as follows:

$$g_r = \mathcal{G}(\{x_j^i \mid x_j^i \in s_r\}), \tag{3}$$

where $\mathcal{G}$ is an average aggregation function used to extract group features from each segmented area.

These features are then incorporated as special tokens in the training process of the MIL model,

$$\{\mathcal{F}_{SAM}(x_j^i; S)\}_{j=1}^n = \{\mathcal{F}(x_j^i)\}_{j=1}^n \oplus \{g_r\}_{r=1}^l. \tag{4}$$

This allows our MIL model to capture the intricate spatial context of the image, thereby improving classification performance. By leveraging the patch features and SAM segmentations extracted during the instance feature extraction phase, we can incorporate SAM's spatial context at two levels during the instance feature aggregation stage.

At the instance level, the abundance of similar and redundant patches in WSIs can lead the MIL model to excessively concentrate on trivial details. This focus on non-essential information can adversely affect the model's ability to generalize and its overall efficiency. **Therefore, we employ a SAM-Guided Group Masking (SG²M) strategy based on spatial context to mask the instance**

features and mitigate class imbalance in WSIs. This process is represented by the following formula:

$$I_k = \{e_p | e_p = M_p \odot \mathcal{F}_{SAM}(x_{kj}), \forall x_j^k \in \mathbf{B_k}\}, \tag{5}$$

where $M_p$ is the patch mask determined by SAM information, $\odot$ represents the Hadamard product, $e_p$ is the masked patch feature embedding, and $I$ represents the instance features.

**At the bag level, SAM-MIL augments the training set by dividing it into additional pseudo-bags and ensuring spatial context consistency across these bags, thereby enhancing its efficacy.** Given the paucity of slide labels, we introduce an approach to create a synthetic diversity of training data. Instance features are randomly allocated into $m$ pseudo bags, as formalized by the following equation:

$$\{P_1^k, P_2^k, \ldots, P_m^k\} = \mathcal{F}_{SAM}(\text{Divide}(\mathbf{B_k}, m)), \tag{6}$$

where $P_1^k, P_2^k, \ldots, P_m^k$ are the feature sets corresponding to each pseudo bag.

We utilize the corresponding SAM segmentation information at both levels to enhance model optimization, and we provide a detailed description of the implementation for each component of this approach below.

## 3.3 SAM-Guided Group Masking Strategy

In the Instance-Level, we introduce a large-scale instance masking strategy, named SAM-Guided Group Masking (SG²M), which is designed to filter out instances that significantly impact the final classification task. Given the presence of numerous similar and redundant instances within WSIs, this repetitive information may cause the MIL model to excessively focus on unimportant details, thereby compromising the model's generalization ability and efficiency. Consequently, the goal of the masking strategy is to diminish the influence of these unimportant instances, ensuring that the model focuses on critical areas. To eliminate the excess of redundant information in the data through masking, we designed an instance-level masking strategy as depicted in Figure 3.

Random Masking Strategy: As demonstrated in Figure 3 (a) and (b), we introduce two parallel types of random masking methods. The first is the "full-feature random mask," which randomly masks the entire feature set to diminish the influence of certain instances, thus reducing redundant information. The second strategy is the "non-group feature random mask," which selectively masks only the features of ordinary instances not included in the group features. This approach strives to preserve the important structural information identified by SAM.

**SAM-Guided Group Masking Strategy**: Since the redundancy degree varies among different types of patches in a slide, employing direct fair masking without consideration can exacerbate the scarcity of already scarce patches. Consequently, we group all patches based on the spatial context provided by SAM, assigning each group a dynamic mask ratio determined by the area information as assessed by SAM. Figure 3 (c) showcases a more refined masking method. Firstly, instances are classified into segment groups $\{G_1, G_2, \ldots, G_z\}$ based on the segmented areas determined by SAM. Each instance $p_i$ is assigned to a segment group $G_k$ according to the category of segmentation it falls under, with the assignment $p_i \rightarrow G_k$ indicating that $k$ is the segment category index of $p_i$.

For each segment group $G_k$, we define $A_{G_k}$ as the area of the group category $k$. Instead of using a direct ratio, we process $A_{G_k}$ through an adjusted sigmoid function to calculate a normalized ratio $R_{G_k}$ for each group,

$$R_{G_k} = \sigma_{\text{adj}}(A_{G_k}) = \frac{1}{1 + e^{-(a \cdot A_{G_k} + b)}}, \tag{7}$$

where $\sigma_{\text{adj}}$ is the adjusted sigmoid function with slope $a$ and centerpoint shift $b$, parameters that control the steepness and the horizontal shift of the sigmoid curve, respectively. The normalized ratio $R_{G_k}$ is then multiplied by the target mask ratio provided as input $(MR_{\text{target}})$, to calculate the final mask ratio for each segment group $MR_{G_k}$,

$$MR_{G_k} = MR_{\text{target}} \cdot R_{G_k}. \tag{8}$$

This ratio defines the extent to which features in each segment group will be masked. Each segment group's mask is calculated and then aggregated to create a composite mask $M_{\text{comp}}$ for all instances,

$$M_{\text{comp}} = \bigcup_{k=1}^{z} M_{G_k}. \tag{9}$$

Finally, the composite mask $M_{\text{comp}}$ is applied across all instances to produce the masked feature set,

$$I_k = Mask(B_k, M_{\text{comp}}), \tag{10}$$

where $Mask(\cdot, \cdot)$ is performing mask operations on instances.

## 3.4 Pseudo-Bag & Consistency Loss

By dividing the instance features from the WSIs into multiple pseudo bags, we effectively amplify our training dataset. Each pseudo-bag inherits the label of its originating slide, thus preserving the ground truth in the expanded dataset. This division into pseudo-bags maximizes the use of available data, reduces model overfitting, and enhances the learning potential of the model. These pseudo-bags are then individually passed through the MIL model for optimization. As illustrated in Figure 4, the mean training loss

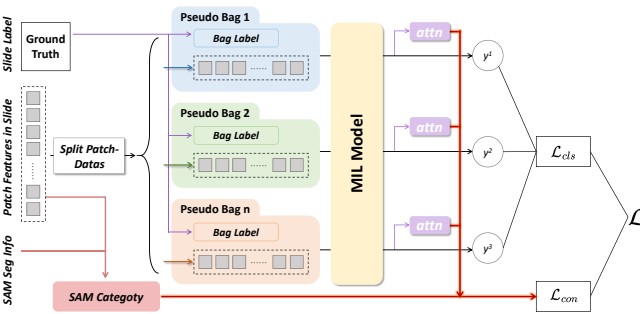

**Figure 4: Illustration of Pseudo-Bag Loss & Consistency Loss.**

across all pseudo bags is computed and utilized as the aggregate loss for model training, encapsulated by the following equation:

$$\mathcal{L}_{\text{cls}}^p = \frac{1}{n} \sum_{i=1}^{n} \mathcal{L}_{\text{MIL}}(Y, C_{SAM}(\mathcal{A}_{SAM}(P_i))), \tag{11}$$

where $\mathcal{L}_{\text{MIL}}$ represents the loss function of the MIL model, $P_i$ represents the instance features in the $i$-th pseudo bag, and $Y$ the inherited label from the WSI.

In addition to the classification loss, we incorporate a consistency constraint based on the segmentation category information derived from SAM. This constraint is based on the premise that global attention weights within the same segmentation category ought to be more similar, thereby enhancing the model's interpretability and its fidelity to the underlying pathology. The consistency loss calculation is as follows:

$$\mathcal{L}_{\text{con}} = \sum_{i=1}^{N} \sum_{j=1}^{N} \text{sim}(attn_i, attn_j) \times (1 - \delta(s_i, s_j)), \tag{12}$$

where sim represents a similarity function between the attention weights $attn_i$ and $attn_j$, and $\delta$ is a Kronecker delta function that equals 1 if the segmentation categories $s_i$ and $s_j$ are the same, and 0 otherwise. The attention weights are derived from the MIL model's attention mechanism, and $s_i$, $s_j$ are the segmentation categories from SAM.

## 3.5 SAM-based MIL

The losses at both the instance level and the bag level, along with the consistency loss, are aggregated for overall model optimization, ensuring that the attention mechanism is not only guided by the MIL classification objective but also respects the intrinsic histological patterns identified by SAM,

$$\{\hat{\mathcal{M}_{SAM}}\} \leftarrow \arg\min \mathcal{L} = \mathcal{L}_{cls} + \alpha \mathcal{L}_{cls}^p + \beta \mathcal{L}_{con}, \tag{13}$$

where $\alpha$ and $\beta$ are scaling factors.

The MIL framework implemented in this work builds upon the widely used AB-MIL [17] architecture prevalent in the WSI domain.

# 4 EXPERIMENTS AND RESULTS

## 4.1 WSI Preprocessing

In order to integrate the segmentation of SAM into the preprocessing workflow of WSIs, it is necessary to adapt the traditional preprocessing protocols. We adopt the data preprocessing protocol

| Method | CAMELYON-16 | | | TCGA Lung Cancer | | |
|---|---|---|---|---|---|---|
| | Accuracy | AUC | F1-score | Accuracy | AUC | F1-score |
| Max-pooling | 78.95±2.28 | 81.28±3.74 | 71.06±2.59 | 87.86±2.59 | 93.89±1.42 | 87.81±2.65 |
| Mean-pooling | 76.69±0.20 | 80.07±0.78 | 70.41±0.16 | 87.48±2.41 | 92.67±2.31 | 87.44±2.43 |
| AB-MIL [17] | 90.06±0.60 | 94.00±0.83 | 87.40±1.05 | 89.67±2.31 | 94.39±1.71 | 89.61±2.32 |
| DSMIL [20] | 90.17±1.02 | 94.57±0.40 | 87.65±1.18 | 89.38±2.69 | 95.03±1.57 | 89.38±2.69 |
| CLAM-SB [27] | 90.31±0.12 | 94.65±0.30 | 87.89±0.59 | 89.29±2.61 | 94.53±1.79 | 89.21±2.65 |
| CLAM-MB [27] | 90.14±0.85 | 94.70±0.76 | 88.10±0.63 | 89.29±2.99 | 94.42±1.93 | 89.24±3.02 |
| TransMIL [34] | 89.22±2.32 | 93.51±2.13 | 85.10±4.33 | 89.10±1.81 | 94.71±1.38 | 89.07±1.81 |
| DTFD-MIL [46] | 90.22±0.36 | 95.15±0.14 | 87.62±0.59 | 90.71±1.63 | 95.39±1.48 | 90.67±1.63 |
| IBMIL* [23] | 91.23±0.41 | 94.80±1.03 | 88.80±0.89 | 89.38±2.42 | 94.59±1.56 | 89.35±2.42 |
| MHIM-MIL* [37] | 90.73±2.61 | 95.72±1.77 | 88.98±2.88 | 90.99±2.27 | 95.77±1.42 | 90.79±2.37 |
| **SAM-MIL** | **91.28±1.94** | **96.08±1.32** | **89.36±2.31** | **91.50±2.27** | **96.01±1.24** | **91.42±2.23** |

**Table 1: The performance of different MIL approaches on CAMELYON-16 (C16) and TCGA Lung Cancer (TCGA). The highest performance is in bold, and the second-best performance is underlined. The Accuracy and F1-score are determined by the optimal threshold. (*MHIM-MIL and IBMIL are two-stage methods that require additional pre-training or clustering operations.)**

of CLAM [27]. The preprocessing workflow of WSIs comprises three stages: Foreground Segmentation, Patching & SAM Segmentation, and Feature & SAM Info Extraction.

**Foreground Segmentation**: This stage is performed strictly in accordance with the CLAM procedure, achieving automatic segmentation of tissue regions, closure of minor gaps and holes, storage of contours, and creation of analysis files.

**Patching & SAM Segmentation**: After completing foreground segmentation, for each slide, our algorithm meticulously crops patches of size $512 \times 512$ from within the segmented foreground contours. It utilizes the HDF5 hierarchical structure to store the image patches along with their coordinates and the slide metadata in a stacked data format. Moreover, each slide is subject to segmentation within the segmented foreground using SAM, with the segmentation information being saved in HDF5 format.

**Feature & SAM Info Extraction**: For each slide, we employ deep convolutional neural networks (the ResNet50 model pretrained on ImageNet, or other feature extractors) to compute local feature representations for each image patch, transforming each $512 \times 512$ patch into a 1024-dimensional feature vector. In addition to the original feature extraction, utilizing the SAM segmentation information saved in the previous stage, a global feature representing each segmentation category is extracted for regions within each category, termed group features, and this information is explicitly annotated and saved in the feature file. Furthermore, the area information of each patch within the SAM segmentation is recorded to provide guidance for subsequent training.

In addition to employing the original process for preprocessing WSIs, we offer a method to generate the aforementioned SAM-guided feature files and related information from previously extracted feature files, reducing the time spent on redundant feature extraction. Please check **Supplementary Material** for more details.

### 4.2 Datasets and Evaluation Metrics

We use **CAMELYON-16 [2]** (C16), and **TCGA-NSCLC** to evaluate the performance on diagnosis and sub-typing tasks. For details on the dataset, please refer to the **Supplementary Material**.

Based on previous work [27, 34], model performance is evaluated using accuracy, area under the curve (AUC), and F1 score. AUC is the primary performance metric in binary classification tasks, and only the AUC and F1 score are reported in the ablation experiments. We adopt ResNet50 [13] pre-trained with ImageNet-1k as the feature extractors. **Supplementary Material** offers more details.

### 4.3 Performance Comparison

We mainly compare with AB-MIL [17], DSMIL [20], CLAM-SB [27], CLAM-MB [27], TransMIL [34], DTFD-MIL [46], IBMIL [23], and MHIM-MIL [37]. In addition, we compared two traditional MIL pooling operations, Max-pooling and Mean-pooling. Due to differences in the datasets, the results of all other methods were reproduced using the provided official code under identical settings.

As demonstrated in Table 1, most of the MIL models in WSI classification tasks focus on how to better utilize local patch features for learning, and many effective models have been designed to improve performance. For instance, MHIM [37] enhances the training for student models by mining hard examples and applying masking to examples based on attention scores during the training process. However, none of the previous work explicitly introduces spatial context; rather, all modeling relies solely on local patch features. The model focuses only on the visual features within each patch obtained through patching, potentially leading to a decrease in performance. Our proposed SAM-MIL explicitly incorporates spatial context and achieves significant performance improvements (96.08% AUC on CAMELYON-16 and 96.01% AUC on TCGA) in both datasets by leveraging segmentation prior information from SAM to guide the training. The performance surpasses that of both MHIM [37] and IBMIL [23], which require two-stage training, underscoring the importance of spatial context information for model training. This not only demonstrates that the direct introduction of spatial context has a positive effect on model training, but also reflects that our designed model is capable of effectively utilizing the additional spatial context to aid in model training.

### 4.4 Ablation Study

| Module | CAMELYON-16 | | TCGA | |
|---|---|---|---|---|
| | AUC | F1 Score | AUC | F1 Score |
| Baseline(AB-MIL) | 94.00 | 87.40 | 94.39 | 89.61 |
| +SAM+Instance Level | 96.01 | 89.33 | 95.84 | 91.81 |
| +SAM+Bag Level | 95.69 | 89.73 | 95.78 | 91.71 |
| +SAM+Instance & Bag | **96.08** | **89.36** | **96.01** | **91.42** |

Table 2: The effect of different components in SAM-MIL with two MIL models.

| Strategy | CAMELYON-16 | | TCGA | |
|---|---|---|---|---|
| | AUC | F1 Score | AUC | F1 Score |
| Baseline(AB-MIL) | 94.54 | 87.44 | 94.27 | 88.69 |
| Random Mask | 95.25 | 89.04 | 95.34 | 90.66 |
| Non-feat. Mask | 95.77 | 89.45 | 95.51 | 91.05 |
| SAM Guided Mask | **96.01** | **89.33** | **95.84** | **91.81** |

Table 3: Comparison between different instance masking strategies.

### 4.4.1 Importance of Different Components.
Table 2 demonstrates the effects of different components in SAM-Guided MIL on two datasets. The experiment's baseline adopts AB-MIL, which is widely used in WSI tasks. First, an instance-level SAM-Guided Group Masking strategy was introduced, utilizing SAM-extracted spatial context to group mask instance features. This strategy resulted in AUC improvements of 2.01% and 1.45% on CAMELYON-16 and TCGA, respectively, thus demonstrating the effectiveness of the instance grouping mask strategy. The performance of different masking strategies in this part will be discussed in detail in Section 4.4.2. Furthermore, the discussion will explore various area calculation functions used in group masking strategies as detailed in Section 4.4.3. Additionally, bag-level pseudo-bag partitioning and a consistency loss based on spatial context were introduced. These approaches also resulted in AUC improvements of 1.69% and 1.39% on CAMELYON-16 and TCGA, respectively, thereby confirming the efficacy of utilizing spatial context at the bag level. After integrating both instance-level and bag-level enhancements, our complete SAM-MIL achieved the best performance (96.08% AUC on CAMELYON-16 and 96.01% AUC on TCGA). Overall, the components of our design effectively utilize the explicitly introduced spatial context at both levels, and the experimental results demonstrate that this explicit spatial context could play a positive guiding role in model training.

### 4.4.2 Impact of Different Masking Strategies.
During the instance group masking phase, one of our key designs is the masking strategy for instances. We introduced three different masking strategies (Random Mask, Non-group feature Mask, and SAM-Guided Group Mask) for instance-level masking. Table 3 shows the performance comparison among different masking strategies across two datasets. First, the experimental results demonstrate that our random masking strategy has already yielded significant improvements. This can be attributed to our extracted group features, which can represent category information well, and the effectiveness of our masking strategy. Furthermore, our proposed SAM-guided group masking strategy achieved the best results in both datasets (96.01%

| Function | CAMELYON-16 | | TCGA | |
|---|---|---|---|---|
| | AUC | F1 Score | AUC | F1 Score |
| Baseline(AB-MIL) | 94.00 | 87.40 | 94.39 | 89.61 |
| Constant | 95.41 | 89.02 | 95.31 | 90.85 |
| Linear Function | 94.23 | 88.32 | 95.06 | 90.86 |
| Adjusted Sigmoid | **96.01** | **89.33** | **95.84** | **91.81** |

Table 4: Comparison between different mask-ratio fuctions.

| Feature | CAMELYON-16 | |
|---|---|---|
| | AUC | F1 Score |
| **TransMIL [34]** | | |
| w/ R50 | 93.51 | 85.10 |
| w/ R50 + Group Feature | 94.66 (+1.15) | 86.27 (+1.17) |
| w/ PLIP | 97.77 | 92.77 |
| w/ PLIP + Group Feature | 97.85 (+0.08) | 92.23 (-0.54) |
| **MHIM-MIL [37]** | | |
| w/ R50 | 96.14 | 89.94 |
| w/ R50 + Group Feature | 96.76 (+0.62) | 91.51 (+1.57) |
| w/ PLIP | 97.79 | 94.13 |
| w/ PLIP + Group Feature | 98.37 (+0.58) | 94.70 (+0.57) |

Table 5: Effect of Group Features in Different Benchmarks.

AUC on CAMELYON-16 and 95.84% AUC on TCGA). This efficacy underscores the value of incorporating spatial contextual information and of applying varied mask ratios across different categories to significantly reduce redundant information. Overall, our proposed instance-level masking strategies have achieved excellent results, confirming their effectiveness across different datasets.

### 4.4.3 Impact of Different Mask-Ratio Function.
Specifically, in the SAM-Guided Group Mask strategy, masks were applied to different categories' groups at varying ratios based on their area. Calculation of the corresponding ratio from the category's area is facilitated by a designated function. In our experiments, three different functions were evaluated: constant, linear, and adjusted sigmoid. The experimental results for these functions are shown in Table 4. First, a fixed ratio showed limited improvements due to data imbalance. A linear function, influenced negatively by extreme values, led to poor performance. The adoption of an adjusted sigmoid function, addressing the drawbacks of the first two methods by adjusting its center-point and slope, resulted in superior performance across datasets.

## 4.5 Effect of SAM Guided Group Feature

Our proposed SAM-Guided Feature Extractor's group features are suitable not only for our designed MIL framework but also for application to various mainstream MIL models, thereby achieving immediate performance improvements. In our experiments, features extracted by ResNet50 [13] and PLIP [16] were processed under the guidance of SAM and validated in the CAMELYON-16 dataset. Besides the previously mentioned baseline AB-MIL, we selected two advanced MIL models [34, 37] to assess the effectiveness of our group features. The corresponding experimental results are displayed in Table 5. The results indicate that incorporating features extracted by SAM led to performance improvements in

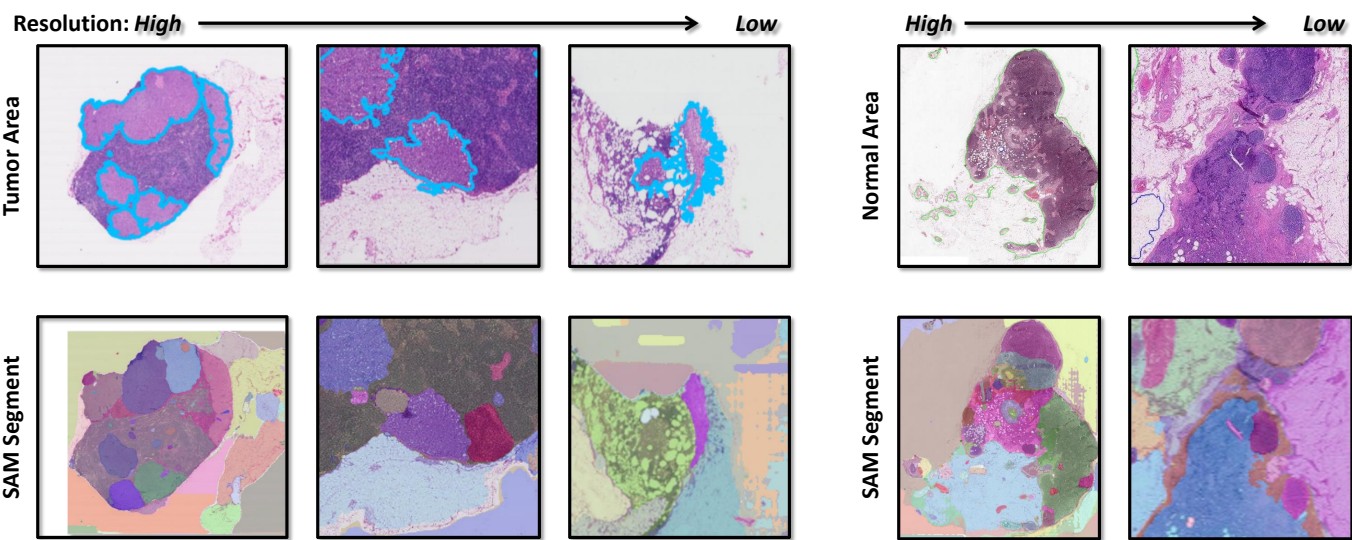

**Figure 5: The figure illustrates the comparison between the slides and their corresponding SAM segmentation results. The first row displays samples of the original slides, including both tumor and normal slides, arranged in descending order by resolution. In the tumor slides, blue lines outline the tumor regions. The arrangement by resolution emphasizes SAM's segmentation performance from macroscopic disease areas to microscopic detail. It is evident that SAM can accurately delineate diseased areas at different scales, and effectively segment normal slides based on visual information.**

both methods. Furthermore, it is observed that, due to the lack of pre-training on medical images, the features extracted using ResNet50 are somewhat inferior in performance compared to those extracted using PLIP. However, after supplementing with group features based on spatial context extraction, the features extracted by ResNet50 demonstrated a greater improvement in performance. This potentially suggests that the group features we proposed provide significant guidance for models lacking prior knowledge. Overall, our SAM-Guided Feature Extractor can serve as a plug-and-play module applicable to various mainstream MIL models.

### 4.6 Visualization

To more intuitively verify the accuracy and effectiveness of the SAM segmentation results, we conducted a comparative visualization of the original tumor and normal slides as shown in Figure 5. The images are arranged from high to low resolution, respectively showcasing SAM's segmentation capabilities from macro to micro levels. From the comparison in the figure, we can observe that the areas outlined in blue lines (tumor regions) in the tumor slides are successfully segmented by SAM, indicating that SAM's segmentation of WSIs is both effective and accurate visually. Similarly, in the normal slides, SAM also segments the WSIs based on visual information, providing additional visual segmentation prior information. On a macro level, whether in tumor or normal slides, SAM is capable of extracting a large number of visually similar regions (which may contain a large number of similar instances leading to redundancy). On a micro level, SAM shows good sensitivity to small local regions, which may be quite important for classification tasks. The experimental results and visualizations also demonstrate that SAM can serve as an effective spatial context extractor for guiding model learning by extracting semantic-free segmentation features,

effectively defining the relations of instances. The potential groups segmented by SAM efficiently distinguish between normal and tumor tissues, ensuring the reliability of the extracted spatial context. In addition to the comparison of SAM's segmentation results, we will provide more visualization information in the **Supplementary Material** to help better understand the model.

## 5 CONCLUSION

In this work, we rethink the impact of spatial contextual information for MIL-based WSI classification tasks. We demonstrate that explicit spatial contextual information is beneficial for MIL classification, and that independent local features may cause the model to neglect the connections between patches and higher-level tissue architecture relations. To address this issue, we employ the visual segmentation foundation model SAM to introduce spatial context by extracting spatial contextual information from image hierarchies in WSIs. Meanwhile, we design multiple components to explicitly introduce the extracted spatial context into the MIL model, thereby guiding the classification of WSIs. Specifically, we develop a SAM-Guided Group Masking strategy to mask instances using spatial contextual information, and extract the representative group features from each category, thereby compensating for the loss of information caused by the masking operation and providing macroscopic feature information. In addition, we introduce spatial context-based consistency loss constraints to enhance the spatial contextual information during pseudo-bags training. The experimental results demonstrate the superiority and versatility of the SAM-MIL framework relative to other methods and further highlight the positive impact of the explicitly introduced spatial context on the MIL model.

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
