# OpenReview forum: "SAM-MIL: A Spatial Contextual Aware Multiple Instance Learning Approach for Whole Slide Image Classification"
_acmmm.org/ACMMM/2024/Conference — MM2024 Poster_

### Official Review · Reviewer_rSyb · 2024-05-24

**Rating:** 1
**Confidence:** 3

**Summary:**

This paper proposes a novel MIL framework to address the problem that current methods usually lead to a significant loss of global spatial context and restrict the model’s focus to merely local features.

**Strengths:**

The figures in this paper are beautiful. Additionally, the paper demonstrates several advantages. First, it significantly improves the performance of multi-instance learning (MIL) models on the CAMELYON-16 and TCGA Lung Cancer datasets by utilizing the Segment Anything Model (SAM) to extract spatial context information. Second, it presents a comprehensive evaluation, demonstrating superior performance compared to mainstream MIL methods, thereby validating the effectiveness of incorporating spatial context in whole slide image (WSI) classification. Third, the enhancement of the training set by creating pseudo-bags while maintaining spatial context consistency is a noteworthy contribution that further improves model robustness and classification accuracy.

**Limitations:**

The experiments were inadequate and poorly analyzed, describing only the general phenomena, but not the causes and reasons behind them. Besides, the dataset scope is relatively narrow, focusing primarily on the CAMELYON-16 and TCGA Lung Cancer datasets, which may not fully represent the diversity of WSI data encountered in clinical settings. The computational complexity of the proposed SAM-MIL framework is high, potentially limiting its practical application in resource-constrained environments. Additionally, the generalizability of SAM to different types of medical images remains unproven, raising concerns about its versatility and applicability in broader medical contexts.

**Suitability:**

1

---

### Official Review · Reviewer_FFqJ · 2024-05-26

**Rating:** 4
**Confidence:** 3

**Summary:**

This paper proposes an MIL framework, named SAM-MIL, that emphasizes spatial contextual awareness and explicitly incorporates spatial context by extracting comprehensive, image-level information. The design of group feature extraction based on spatial context and a SAM-guided group masking strategy are used to mitigate class imbalance issues. A dynamic mask ratio for different segmentation categories is utilized. SAMMIL divides instances to generate additional pseudo-bags, and introduces consistency of spatial context across pseudo-bags to enhance the model’s performance. Experimental results on the CAMELYON-16 and TCGA Lung Cancer datasets demonstrate the proposed SAM-MIL model is effective.

**Strengths:**

1.	Overall, this paper is written clearly.
2.	The topic is interesting and significant.
3.	Many experiments are provided to support the conclusions.

**Limitations:**

1.	In the Title and Abstract, medical images should be mentioned to better understand whole slide image.
2.	This paper emphasizes spatial contextual awareness, but the concept is general and almost all methods in this field perform spatial contextual awareness.
3.	In experiments, many details are missing, such as some settings on patches, bags, and multiple instance learning.
4.	My main concern is whether or not SAM is effective for segmenting medical slide images.

**Suitability:**

2

---

### Official Review · Reviewer_WghJ · 2024-05-28

**Rating:** 4
**Confidence:** 4

**Summary:**

This paper proposes a spatial contextual aware MIL approach (SAM-MIL) for whole slide image classification. Current MIL-based methods usually focus on the local small patches, leading to a significant loss of global contextual spatial information. SAM-MIL explicitly models the spatial information by utilizing SAM and proposes several spatial contextual information mining strategies at both the bag level and instance level.

**Strengths:**

1. The writing and logical structure of this paper are good and easy to read.
2. Improving the MIL's framework performance by utilizing the SAM-guided spatial contextual information is novel in the field of digital pathology.
3. Extensive comparable experiments and ablation studies verify the effectiveness of the proposed SAM-MIL.

**Limitations:**

1. Modeling the spatial context in the WSI to enhance the performance of the MIL framework is a hot topic.  Some latest graph-based MIL-related work is missing [1-4].
2. SAM is pre-trained in the natural image. The specific discussion about why it can be directly used to segment the WSI needs to be analyzed.
3. SAM relies on visual or textual prompts to perform segmentation. However, I couldn't find a description of the prompts used in this paper. How does the author utilize SAM to generate the segmentation mask?
4. There are some other methods to segment the WSI and provide spatial information, such as K-means and SLIC [5]. A comparison and discussion between SAM-based methods and these traditional methods is necessary.
5. In the Camelyon-16 dataset, the baseline results (e.g., DTFD-MIL and TransMIL) match those reported in the paper [6]. Why are the MHIM-MIL results different?
6. A paired t-test statistical analysis (p-value > 0.5) is necessary to evaluate the significance of the performance improvement.
7. Why are there white Spaces in the upper and left sides of the segmentation results in the lower left corner of Figure 5?

Reference:
[1] Hou W, Yu L, Lin C, et al. H^2-MIL: exploring hierarchical representation with heterogeneous multiple instance learning for whole slide image analysis[C]//Proceedings of the AAAI conference on artificial intelligence. 2022, 36(1): 933-941.
[2] Lee Y, Park J H, Oh S, et al. Derivation of prognostic contextual histopathological features from whole-slide images of tumours via graph deep learning[J]. Nature Biomedical Engineering, 2022: 1-15.
[3] Chan T H, Cendra F J, Ma L, et al. Histopathology whole slide image analysis with heterogeneous graph representation learning[C]//Proceedings of the IEEE/CVF Conference on Computer Vision and Pattern Recognition. 2023: 15661-15670.
[4] Li J, Chen Y, Chu H, et al. Dynamic Graph Representation with Knowledge-aware Attention for Histopathology Whole Slide Image Analysis[J]. arXiv preprint arXiv:2403.07719, 2024.
[5] Achanta R, Shaji A, Smith K, et al. SLIC superpixels compared to state-of-the-art superpixel methods[J]. IEEE transactions on pattern analysis and machine intelligence, 2012, 34(11): 2274-2282.
[6] Tang W, Huang S, Zhang X, et al. Multiple instance learning framework with masked hard instance mining for whole slide image classification[C]//Proceedings of the IEEE/CVF International Conference on Computer Vision. 2023: 4078-4087.

**Suitability:**

2

---

### Meta-Review · Area_Chair_q63t · 2024-07-02

**Recommendation:** Accept (Poster)
**Confidence:** 4

**Metareview:**

Overall, all reviewers are satisfied with the response given by the authors, and are glad to see that the quality of the paper has been improved substantially.